# Characterisation of NLRP3 pathway-related neuroinflammation in temporal lobe epilepsy

**Malin S. Pohlentz[1], Philipp Müller[1], Silvia Cases-Cunillera[1], Thoralf Opitz[2], Rainer Surges[3,4], Motaz Hamed[5], Hartmut Vatter[5], Susanne Schoch[1,3], Albert J. Becker[1], Julika Pitsch[1,3] ***

1 Dept. of Neuropathology, Section for Translational Epilepsy Research, University Hospital Bonn, Bonn, Germany, 2 Inst. for Experimental Epileptology and Cognition Research, University Hospital Bonn, Bonn, Germany, 3 Dept. of Epileptology, University Hospital Bonn, Bonn, Germany, 4 Center for Rare Diseases Bonn (ZSEB), University Hospital Bonn, Bonn, Germany, 5 Dept. of Neurosurgery, University Hospital Bonn, Bonn, Germany

☯ These authors contributed equally to this work.

* jpitsch@uni-bonn.de

**Data Availability Statement:** All relevant data are within the paper and its Supporting Information files.

## Abstract

### Objective

Inflammation of brain structures, in particular the hippocampal formation, can induce neuronal degeneration and be associated with increased excitability manifesting as propensity for repetitive seizures. An increase in the abundance of individual proinflammatory molecules including interleukin 1 beta has been observed in brain tissue samples of patients with pharmacoresistant temporal lobe epilepsy (TLE) and corresponding animal models. The NLRP3-inflammasome, a cytosolic protein complex, acts as a key regulator in proinflammatory innate immune signalling. Upon activation, it leads to the release of interleukin 1 beta and inflammation-mediated neurodegeneration. Transient brain insults, like status epilepticus (SE), can render hippocampi chronically hyperexcitable and induce segmental neurodegeneration. The underlying mechanisms are referred to as epileptogenesis. Here, we have tested the hypothesis that distinct NLRP3-dependent transcript and protein signalling dynamics are induced by SE and whether they differ between two classical SE models. We further correlated the association of NLRP3-related transcript abundance with convulsive activity in human TLE hippocampi of patients with and without associated neurodegenerative damage.

### Methods

Hippocampal mRNA- and protein-expression of NLRP3 and associated signalling molecules were analysed longitudinally in pilocarpine- and kainic acid-induced SE TLE mouse models. Complementarily, we studied NLRP3 inflammasome-associated transcript patterns in epileptogenic hippocampi with different damage patterns of pharmacoresistant TLE patients that had undergone epilepsy surgery for seizure relief.

**Funding:** Our work is supported by the Deutsche Forschungsgemeinschaft (to AJB: SFB 1089; to SS: SCHO 820/4-1, SCHO 820/6-1, SCHO 820/7-1, SCHO 820/5-2, SPP1757, SFB1089; to AJB: FOR 2715), Else Kröner-Fresenius-Foundation (Promotionskolleg 'NeuroImmunology' to MP, SS, AJB; 2016_A05 to JP), as well as the BONFOR program of the Medical Faculty, University of Bonn (MP, SS, AJB, JP). The funders had no role in study design, data collection and analysis, decision to publish, or preparation of the manuscript. There was no additional external funding received for this study.

**Competing interests:** The authors have declared that no competing interests exist.

## Results

Pilocarpine- and kainic acid-induced SE elicit distinct hippocampal *Nlrp3*-associated molecular signalling. Transcriptional activation of NLRP3 pathway elements is associated with seizure activity but independent of the particular neuronal damage phenotype in KA-induced and in human TLE hippocampi.

## Significance

These data suggest highly dynamic inflammasome signalling in SE-induced TLE and highlight a vicious cycle associated with seizure activity. Our results provide promising perspectives for the inflammasome signalling pathway as a target for anti-epileptogenic and -convulsive therapeutic strategies. The latter may even applicable to a particularly broad spectrum of TLE patients with currently pharmacoresistant disease.

## Introduction

Temporal lobe epilepsy (TLE) is the most frequent form of focal epilepsy in adults [1] and is often triggered by a transient brain insult or injury. A cascade of molecular, cellular and physiological alterations results in the evolution of chronic recurrent seizures. This development of an epileptic condition and/or the progression of epilepsy after its establishment is referred to as 'epileptogenesis' [2] but the underlying mechanisms have remained unresolved. A better understanding of the pathogenesis is urgently needed, as a fraction of TLE patient does not respond to antiseizure drugs [1]. Surgical removal of the epileptogenic focus can potentially decrease or even abolish seizures in many patients suffering from drug-refractory TLE [3]. The most frequent neuropathological finding in resected tissue of these individuals is hippocampal sclerosis (HS) [3], which is characterised by segmental neuronal cell loss, microgliosis and reactive astrogliosis [4, 5]. This lesion pattern in concert with the development of chronic recurrent seizures is reproduced in Status epilepticus (SE) induced TLE mouse models [6] induced either by the excitotoxic compounds pilocarpine [7] or kainic acid (KA) [8, 9].

Recent data point to an important role of aseptic inflammatory signalling in the development and maintenance of TLE [10, 11]. Blockade of pro-inflammatory signalling molecules represents a promising approach for a disease-modifying treatment of TLE [12–15]. An example of an important key regulator responsible for the release of proinflammatory cytokines is the NOD-like receptor protein 3 (NLRP3). Upon toll-like receptor 4 (TLR4) activation, downstream signalling leads to Nuclear Factor κB (NFκB) dependent expression of both pro-interleukin-1-beta (*pro-Il1b*) and *Nlrp3* mRNA [16]. After transcriptional upregulation, a second danger signal leads to the formation of the NLRP3 inflammasome [17], which comprises NLRP3, apoptosis-associated speck-like protein (ASC) and pro-caspase-1. The inflammasome then cleaves pro-caspase-1, which promotes Interleukin 1 beta (IL1B) activation and release [18].

Inhibition of individual components of the NLRP3 pathway attenuates neuroinflammation and reduces seizure frequency and severity in experimental TLE [12, 15, 19–21]. Molecules involved in NLRP3 signalling have been detected in HS-TLE, in which the hippocampus shows extensive neuronal damage [12, 22–25]. si-RNA mediated knockdown of *Nlrp3* acted neuroprotectively and also attenuated the chronic seizure phenotype in an electrically induced SE model in rats [13]. *NLRP3* knockout reduced neuronal necrosis and apoptosis as well as peripheral blood levels of IL1B one week after pilocarpine-induced SE in mice [12].

SE-induced TLE models have different structural and functional characteristics dependent on the excitotoxic compound (for review see [26]). Here, we have examined, which hippocampal inflammasome-related molecular dynamics emerge from systemic pilocarpine- versus unilaterally suprahippocampal KA-induced SE. Additionally, we investigated whether the extent of neuronal damage affects transcriptional changes of inflammasome-related molecules. In a translational effort, we have further tested, whether differential inflammasome-related molecular signatures occur in chronic TLE hippocampi with epileptogenesis and extensive neuronal damage versus hippocampi from lesion-associated TLE. The latter are characterized by a general lack of epileptogenesis and HS, despite being chronically epileptic. Our results suggest robust and sustained activation of inflammasome-mediated signalling in experimental and human TLE hippocampi independent of a specific damage pattern.

## Methods

### Animals

All animal procedures were planned and performed to minimize pain and suffering and to reduce the number of used animals in accordance with the guidelines of the University Hospital Bonn, Animal-Care-Committee as well as the guidelines approved by the European Directive (2010/63/EU) on the protection of animals used for experimental purposes and ARRIVE guidelines. All mice were housed in a humidity (55 ± 10%) and temperature (22 ± 2°C) controlled environment under a 12-h light–dark-cycle (light cycle 7 am to 7 pm) with water and food ad libitum and nesting material (Nestlets, Ancare, USA). Mice were allowed to adapt to the animal facility at least for seven days before any treatment.

### Induction of chronic epilepsy by suprahippocampal kainic acid injection

As described previously, anaesthetized [16 mg/kg xylazine (Ceva Tiergesundheit, Germany) and 100 mg/kg ketamine, i.p. (Ketamin 10%, WDT, Germany)] male C57Bl6/N mice (Charles River; ~60 days old, weight ≥ 20 g) were used for induction of chronic epilepsy. Briefly, 70 nl kainic acid (20 mM in 0.9% NaCl, Tocris) was applied over 2 min unilaterally above the CA1 region of the left dorsal hippocampus (stereotaxic coordinates in mm relative to Bregma: -2 anterioposterior, -1.5 mediolateral, and -1.4 dorsoventral) using a Nanofil syringe (WPI) operated by a micropump (WPI) [27]. Control (non-SE) mice were injected in the same manner with 70 nl 0.9% NaCl. After injection the syringe was left for additional 2 min to avoid reflux. All operated mice received analgesic treatment before and once per day for 3 days post-surgery (5 mg/kg Ketoprofen, s.c.; Gabrilen, mibe). With this method the incidence of inducing SE is 100% without any mortality [27]. All animals exhibit chronic behavioural focal to bilateral tonic-clonic seizures (10d, 28d group).

### Induction of chronic epilepsy by systemic pilocarpine injection

To induce SE, animals (male C57Bl6/N mice; Charles River; ~60 days old, weight ≥ 20 g) were injected subcutaneously with 335 mg/kg pilocarpine hydrochloride (Sigma), 20 min after pretreatment with subcutaneous injection of 1 mg/kg scopolamine methyl nitrate (Sigma) as described in previous publications [28, 29]. Forty min after SE onset, animals received an injection of diazepam (4 mg/kg, s.c.; Ratiopharm). Control (non-SE) animals were treated identically but received saline instead of pilocarpine. Behavioural SE was clearly identified using a modified seizure scheme sustained continuous convulsions with postural loss designated as SE [28, 29]. Among pilocarpine injected animals, only those that developed SE (SE-

experienced) were further used for analysis. All SE-experienced animals showed behavioural focal to bilateral tonic-clonic seizures at later model stages (5d, 10d, 28d group).

## Telemetric video/EEG monitoring

Semiological aspects of were analysed in male C57Bl6/N mice (Charles River; ~60 days old, weight $\geq$ 20 g) for 28 days after KA- or pilocarpine-induced SE by using 24/7 telemetric EEG/ video monitoring. EEG implantation, postoperative treatment, continuous telemetric EEG-/ video-monitoring, and behavioral seizure classification were previously described in detail [27, 29]. Two weeks before pilocarpine injection or directly after KA injection, a cortical ECoG electrode was implanted (in mm relative to Bregma: -2 AP, -1.5 ML) with subcutaneously placed transmitters (ETA-F20, DSI). A stainless-steel screw in contact with the cerebellar cortex of the simple lobe at the midline (in mm relative to Bregma: -6 AP, 0 ML, 0 DV) was used to fix the reference electrode. Appropriate placement of the electrodes was assessed histologically in all mice at the end of the experiment. Continuous EEG recording with a sampling rate of 1 kHz was started directly after electrode implantation and performed and analysed using NeuroScore v3.2 (DSI) software. From concurrent video recordings, seizures were classified as described before [27, 29].

## Murine mRNA isolation, real-time RT-PCR quantification, histopathological analysis, cell quantification

mRNA was isolated from microdissected hippocampal CA1 region at different time points after induced SE using magnetic beads (mRNA Direct™ Micro Kit, Invitrogen) followed by cDNA synthesis (RevertAid H Minus First Strand cDNA Synthesis Kit, Thermo Scientific) as described before [27]. Briefly, transcripts of different genes were quantified by RT-PCR using CFX384 Touch Real-Time PCR Detection System (Biorad) and determined with the $\Delta\Delta C_t$ method. PCR samples for quantification of ASC and NFκB2 contained 3 μl Maxima Probe/ ROX qPCR Master Mix (2x) (Thermo Scientific), 0.3 μl of TaqMan™ Gene Expression Assay (Thermo Scientific, Mm00479807_m1, Hs01547324_gH), 1.7 μl RNase free water and 2 μl cDNA. All other PCR samples contained 3.125 μl Maxima SYBR Green/ROX qPCR Master Mix (2x) (Thermo Scientific, K0223), 0.3 μM of each oligonucleotide primer (*Nlrp3*: FW: AATGCCCTTGGAGACACAGG and RV: ATTCCAGCAGCTGTGTGAGG; *Casp1*: FW: GGACCCTCAAGTTTTGCCGCT and RV: ATGAGGGCAAGACGTGTACG; *Aif1*: FW: TCAGAAT GATGCTGGGCAAG and RV: GACCAGTTGGCCTCTTGTGT; *Gfap*: FW: AGAAAACCGCATCA CCATTC and RV: TCTTGAGGTGGCCTTCTGAC; *Il1b*: FW: CACTACAGGCTCCGAGATGA and RV: TTTGTCGTTGCTTGGTTCTC; TLR4: FW: CCAATTTTTCAGAACTTCAGTGG and RV: AGAGGTGGTGTAAGCCATGC; *β-actin*: FW: ACCGTGAAAAGATGACCCAGA and RV: ATGGG CACAGTGTGGGTGA), 1.5 μl cDNA and 1.5 μl of RNase free water. RT-PCR was performed using the following steps: 2 min at 50˚C, 10 min at 95˚C, then 40 cycles of 15 s at 95˚C, 30 s at 60˚C and 30 s at 72˚C. For KA-injected animals, the data of left and right CA1 region were pooled because no significant differences were found between both hemispheres (S1A Fig).

For immunohistochemistry, mice were decapitated under isoflurane anaesthesia (Forene®, Abbott, Germany). Brains were removed quickly, fixed in 4% PFA overnight and embedded in paraffin. Coronal sections (4 μm thick) were used for GFAP (glial fibrillary protein; to label reactive astrogliosis, NeuroMab, 1:250), NeuN (Neuronal nuclei; to label neuronal somata, Millipore, 1:500), AIF1 (allograft inflammatory factor 1; to label activated microglia, Wako, 1:1000), IL1B (Interleukin 1 beta, abcam, 1:100) as described previously [27]. Briefly, paraffin sections were deparaffinized and washed in distilled water. Antigen retrieval mediated by citric acid (10mM, pH 6,0) was performed at 90˚C for 8 min. After washing, brain sections were

blocked for 2 h at 37˚C in PBS buffer containing 10% normal goat serum, 1% fetal calf serum, and incubated with primary antibodies over night at 4˚C. After washing, slides were incubated with respective secondary antibodies (Life technologies, 1:100) and DAPI (4′,6-Diamidino-2-phenylindole dihydrochloride, Sigma, 1:100.000) in blocking buffer, washed and mounted with Mowiol 4–88 (Roth).

For semi-quantitative analysis of microglia and astrocytes, DAB (3,3′-Diamino-benzidin) staining was used. After blocking with 5% normal goat serum diluted in PBS for 1 h at 37˚C, slices were washed and incubated with GFAP (Sigma, 1:500) and AIF1 (Wako, 1:1000) overnight at 4˚C. After washing, slices were incubated with respective biotinylated secondary antibodies (Vector Laboratories, USA, 1:200) in blocking buffer and incubated for 2 h at 37˚C. Slides were again washed and further processed according to manufacturer's protocol (Sigma-fast, Merck, Germany) and mounted with Mowiol 4–88 (Roth).

All non-fluorescent digital images were captured with a microscope (BZ-X, Keyence, Japan) and fluorescent images with a confocal microscope (Nikon, Eclipse T*i*). For cell quantification, Fiji (ImageJ) software was used to apply colour deconvolution to change RGB images to DAB and a global threshold was applied to the DAB channel to get a binary image. The resultant data was extracted as area fraction (ranging from 0 to 100). Three sections from each mouse (between -1.6 mm and -2.1 mm anterio-posterior relative to Bregma) and within each section, three adjacent fields of CA1 were quantified.

## Human TLE patients, mRNA sequencing and histopathological analysis

Gene expression analyses were applied on human hippocampal biopsy tissue from patients with hippocampal sclerosis (n = 78) versus patients with lesion-associated chronic TLE (involving low-grade neoplasms / dysplasia; n = 34), who underwent surgical treatment in the Epilepsy Surgery Program at the University Hospital Bonn due to pharmacoresistance [30]. The general clinical data are now summarized in **S1 Table**. In all patients, presurgical evaluation using a combination of non-invasive and invasive procedures revealed that seizures originated in the mesial temporal lobe [31]. For each case included in the present study, hippocampal tissue samples were available for neuropathologic examination. The HS group was clearly characterized by segmental neuronal cell loss and concomitant astrogliosis and microglial activation. The hippocampi in the control group showed no segmental neuronal cell loss neuropathologically but exhibited astrogliosis and microglial activation and were therefore consistent with lesions such as cortical dysplasia or epilepsy-associated tumors. In each case, the diagnosis was made by an experienced neuropathologist (AJB) according to international criteria [32, 33]. All procedures were conducted in accordance with the Declaration of Helsinki and approved by the Ethics Committee of the University Hospital Bonn (222/16). Informed written consent was obtained from all patients. mRNA analyses for *Nlrp3*, *Casp1*, *Tlr4*, *NfkB2*, *Il1b*, *Il18*, *Gfap* and *Aif1* were carried out analogous to a procedure described elsewhere in detail [34]. Briefly, RNA from hippocampal biopsies served to generate 750 ng cRNA used for hybridization on Human HT-12 v3 Expression BeadChips with Illumina Direct Hybridization Assay Kit (Illumina, San Diego, CA) according to standard procedures. We extracted data for our genes of interest analysed by Illumina's GenomeStudio Gene Expression Module and normalized using Illumina BeadStudio software suite by quantile normalization with background subtraction.

For histological analysis, human surgical brain tissue was fixed with formaldehyde overnight, embedded into paraffin. 4 μm sections were used for both, hematoxylin-eosin (H&E) staining and immunohistochemistry by using standard protocols [35]. Briefly, paraffin sections were de-paraffinized in xylene for 10 min followed by decreasing alcohol series (100–

50%) for 2 min each. After washing, slides were subjected to a citric acid antigen retrieval procedure, followed by two washing steps. Subsequently, brain sections were rinsed for 2 h at 37°C in PBS blocking buffer (10% normal goat serum), and incubated overnight at RT with antibodies against glial fibrillary acidic protein (GFAP, Z0334, Dako, 1:100), neuronal nuclei (NeuN, MAB377, Milipore, 1:1000), AIF1 (Wako, 1:1000), or NLRP3 (abcam, ab214185, 1:100) in PBS-blocking buffer. DAB (3,3′-Diamino-benzidin) staining was done as on murine slices (see section above). For fluorescent-immunohistochemistry, after washing slides were incubated with secondary antibody (Alexa Fluor®568, Life technologies, Germany, 1:200) and DAPI (1:100.000, Sigma, Germany) and mounted with Mowiol 4–88 (Roth, Germany). The NLRP expression pattern was evaluated by using morphological analysis.

## Statistical analysis

Experiments were conducted in a randomized and blinded fashion. Prism 9 software (Graph-Pad Software) was used for statistical analysis. Sample size ($n$) per experiment was calculated using power analysis, with parameters set within the accuracy of the respective experiment. To analyse the impact of SE on the expression level of NLRP-related genes, two-way ANOVA was used with SE and time after SE being the two categorical independent variables. Sidak's multiple comparison test was used to pinpoint significant differences between control and SE group at certain time points. Mann Whitney U-test was used to analyse NLRP-related gene expression between lesion and hippocampal sclerosis sites in human specimens (for detailed statistical analysis see S2 Table). For correlation analysis we used the median of the summarized seizure frequency of three consecutive days before mRNA analysis for every animal (day 3: seizure sum of day 1–3; day 5: sum of day 3–5; day10: seizure sum of day 8–10; day 28: sum of day 26–28) after KA (n = 21) and pilocarpine-induced SE (n = 11). The median seizure frequency was correlated with average gene expression using simple linear regression. Statistical outliers were identified by using Grubb's test (GraphPad) and excluded from analysis. Differences between means were considered significant at p<0.05. All results are plotted as mean ± SD. Negative error bars crossing the x-axis limit were clipped.

## Results

### Transcription of NLRP3 and associated inflammasome signalling molecules is activated in a model- and time-specific manner

Transcript dynamics are important in order to trigger an inflammasome response to an insulting stimulus. Therefore, we first determined the time course of gene expression of individual molecules of the NLRP3 pathway (Fig 1A) during four weeks following subcutaneous pilocarpine or unilateral suprahippocampal KA injection. After pilocarpine-induced SE, *Nlrp3* expression was increased during the early stage with the highest expression after 10 days (for detailed statistical analysis see S2 Table). A similar pattern was observed for other inflammasome components, namely for transcripts encoding the adaptor-molecule *Asc* and the effector-molecule *Casp1*. Transcription of genes involved in the signalling cascade prior to inflammasome formation including *Tlr4* and *Nfkb2*, was also significantly augmented at 72 hours and 10 days after pilocarpine induced SE. During the later epileptic state (28 days post SE) mRNA expression was indistinguishable from control levels (Fig 1B). In contrast to pilocarpine-induced SE, after KA-induced SE, an increase of mRNA levels was not only observed during the early disease progression (72 hours and 5 days post SE) but also in chronic TLE stages. However, 10 days after KA-induced SE mRNA levels returned back to basal (Fig 1C). There

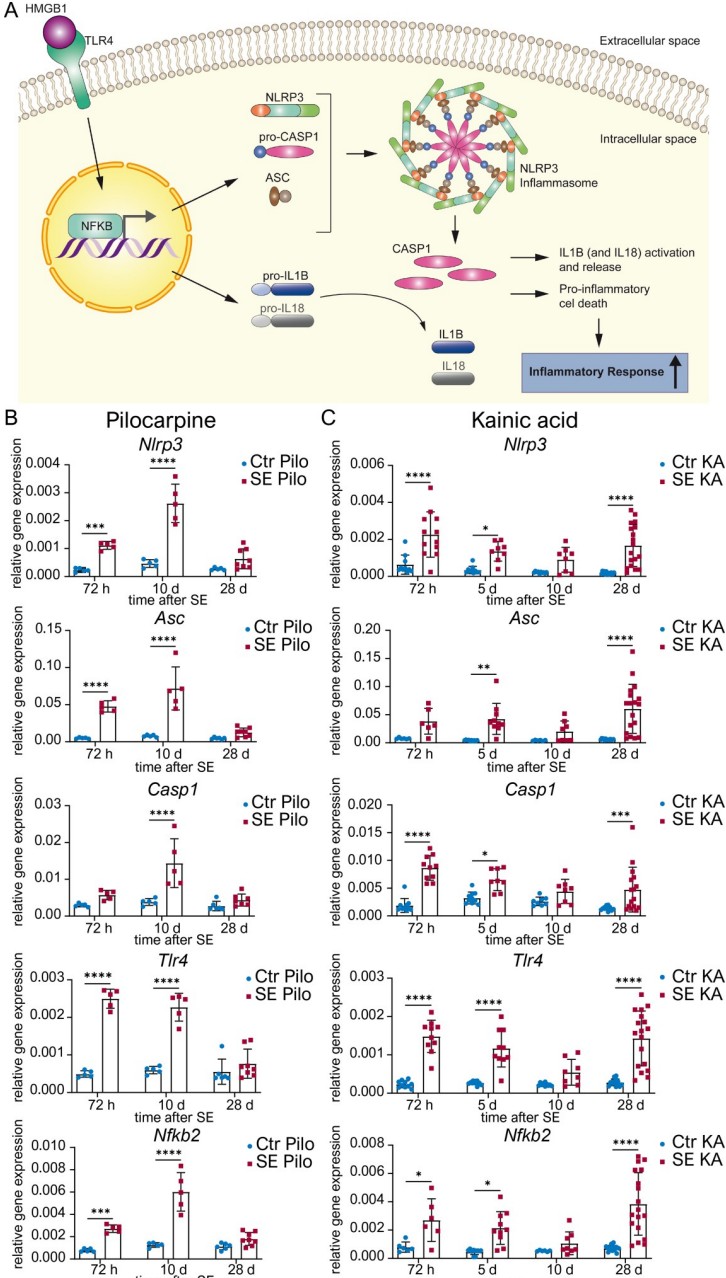

**Fig 1. mRNA expression of NLRP3 and associated signalling molecules are differently regulated after SE elicited by pilocarpine or kainic acid.** (**A**) Schematic representation of the NLRP3 pathway: After activation of Toll-like receptor (TLR) 4 by High-Mobility-Group-Protein (HMG) B1, nuclear factor 'kappa-light-chain-enhancer' of activated B-cells (NFKB) 2 is activated and translocates to the nucleus. Here, it enhances transcription of NOD-like receptor protein (*Nlrp*) 3 and Interleukins. Subsequently, NLRP3, Apoptosis-like-speck protein (ASC) and an inactive precursor of cysteine-dependent aspartate-specific Protease (CASP) 1 assemble to form the NLRP3 inflammasome. Upon activation of the inflammasome, pro-CASP1 is cleaved and activated, which then leads to cleavage of interleukin (IL) 1B and 18, as well as proinflammatory cell death. (**B**) Relative mRNA levels (ΔΔC$_t$ method) of inflammasomal associated genes are analysed 72 h (n ≥ 5), 10 d (n ≥ 5) and 28 d (n ≥ 6) after pilocarpine-induced status epilepticus (SE Pilo) as well as (**C**) 72 h (n ≥ 4), 5 d (n ≥ 5), 10 d (n ≥ 4) and 28 d (n ≥ 6) after kainic acid-induced SE in hippocampal CA1 (SE KA) compared to non-SE controls (Ctr Pilo/Ctr KA). Relative mRNA levels of NOD-like receptor protein (*Nlrp*) 3 and other inflammasomal associated genes such as *Asc*, *Casp1*, *Tlr4*, and *Nfkb2* are quantified with ΔΔC$_t$ method normalized to the ubiquitously expressed (encoding proteins that are required for all cell types) housekeeping gene *β-actin* after SE. Data is shown as mean ± SD and analysed with 2way ANOVA followed by Sidak's post hoc test. Asterisks indicate significant differences between groups (control vs. SE): *p < 0.05, **p < 0.01, ***p < 0.001, ****p < 0.0001. Detailed statistical values are found in **S2 Table**.

was no significant difference in mRNA expression level between the ipsilateral extensively damaged (KA-injected) and the contralateral side of the hippocampus (**S1A Fig**).

With respect to downstream effects of inflammasome activation in both experimental models, interleukin 1 mRNA and protein expression were analysed. Increased transcription of Interleukin 1 beta (*Il1b*) was only observed 10 days after pilocarpine induced SE compared to expression in the CA1 region of non-SE animals (**Fig 2A,** for detailed statistical analysis see **S2 Table**). In contrast, a significant decrease of Interleukin 18 (*Il18)* mRNA levels was found at 72 hours post SE (**Fig 2B**). A strong increase in IL1B levels was observed 72 hours after pilocarpine-induced SE in a clustered subgroup of CA1 neurons (**Fig 2C**). Additionally, few non-neuronal cells showed IL1B immunoreactivity. Ten days after SE, only a weak IL1B expression was detected in the remaining CA1 pyramidal neurons. However, in areas with extensive neuronal degeneration, non-neuronal cells were strongly positive for IL1B at 28 days after SE including astrocytes (**Fig 2C and 2D**).

In contrast, a significant increase of *Il1b* mRNA was observed at all time points investigated after KA-induced SE (**Fig 3A**, for detailed statistical analysis see **S2 Table**), whereas

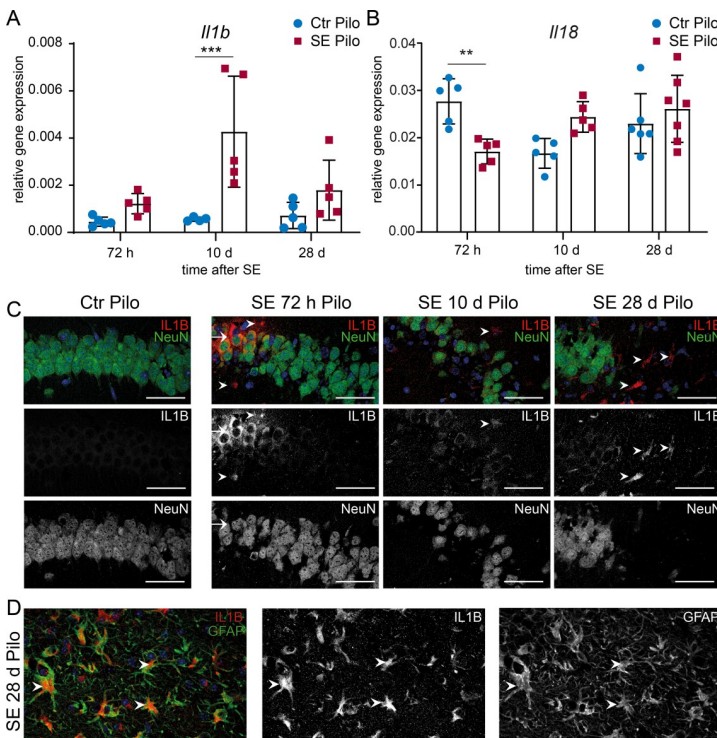

**Fig 2. mRNA and protein expression levels of Interleukin 1 beta (IL1B) are transiently augmented after pilocarpine-induced SE in hippocampal CA1 region.** Relative mRNA levels ($\Delta\Delta C_t$ method) of (**A**) *Il1b and* (**B**) *Il18* are analysed 72 h (n ≥ 5), 10 d (n ≥ 5) and 28 d (n ≥ 6) after pilocarpine-induced SE compared to non-SE control animals (Ctr Pilo) with 2way ANOVA followed by Sidak's post hoc test. Asterisks indicate significant differences between groups (control vs. SE): *p < 0.05, **p < 0.01, ***p < 0.001, ****p < 0.0001. Detailed statistical values are found in **S2 Table**. The ubiquitously expressed *β-actin* is used as reference gene. (**C**) Representative immunohistochemistry of the hippocampal CA1 region shows 72 h after pilocarpine-induced SE a segmental expression of IL1B (red) exclusively in the CA1 pyramidal cell layer (NeuN positive neurons = green). Some non-neuronal cells are faintly IL1B positive 10 days post-SE (arrowheads). However, 28 d after SE numerous strongly IL1B positive cells are visible (arrowheads), whereas no IL1B expression is detected in non-SE control animals (leftmost, Ctrl Pilo). Scale bars: 50 μm. (**D**) A strong co-localization with GFAP within the former pyramidal cell layer indicating IL1B positive astrocytes 28 days after SE induction. Scale bars: 50 μm. 4′,6-diamidino-2-phenylindole (DAPI) for nucleic acid counterstain (blue).

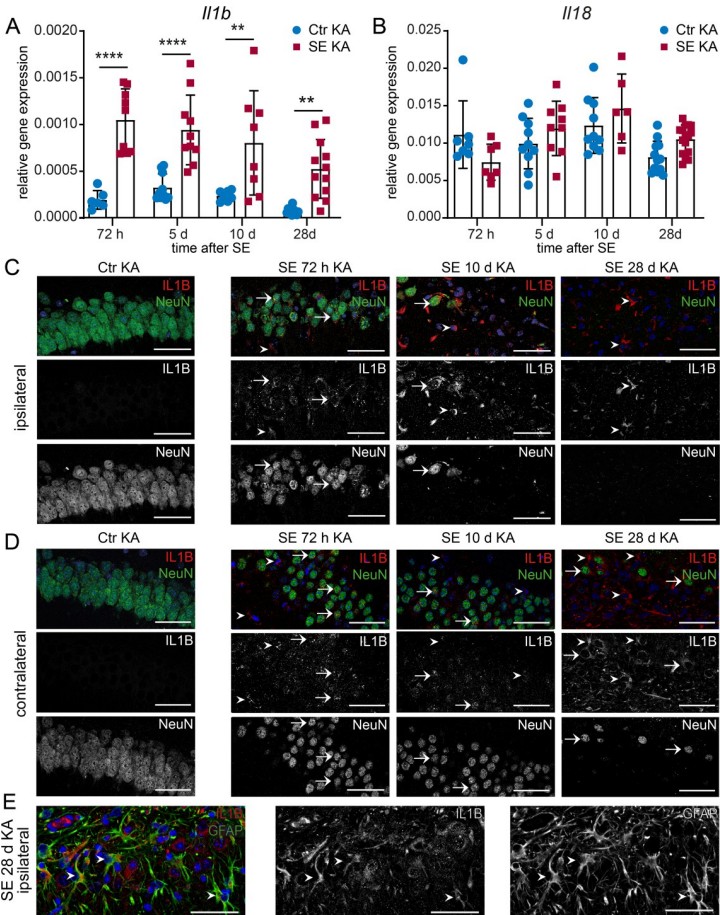

**Fig 3. mRNA and protein expression levels of Interleukin 1 beta (IL1B) are augmented in the CA1 region over a period of 4 weeks after kainic acid-induced SE.** (**A**) *IL1b* and (**B**) *Il18* mRNA levels ($\Delta\Delta C_t$ method) are determined 72 h (n ≥ 4), 5 d (n ≥ 5), 10 d (n ≥ 4) and 28 d (n ≥ 6) after kainic acid-induced SE compared to non-SE control animals (Ctr KA) with 2way ANOVA followed by Sidak's post hoc test. Asterisks indicate significant differences between groups (control vs. SE): *p < 0.05, **p < 0.01, ***p < 0.001, ****p < 0.0001. Detailed statistical values are found in **S2 Table**. The ubiquitously expressed *β-actin* is used as housekeeping gene. Immunohistochemistry of (**C**) ipsilateral (KA-injected) as well as (**D**) contralateral hemisphere shows single IL1B granula (red) within NeuN-postive cells (green) at 72 h post-SE (arrows). Additionally, very few non-neuronal IL1B positive cells are found (arrowheads). 10 d post-SE IL1B immunoreactivity is detected in neurons (arrows) and non-neuronal cells expressing IL1B, observed mainly in areas with vast neuronal cell loss. At 28 d post-SE, a high number of cells are IL1B positive (arrowheads) including also remaining NeuN positive cells in the contralateral hemisphere (arrows). In non-SE control animals (Ctrl KA), expression of IL1B is not detected (leftmost). Scale bars: 50 μm. DAPI for nucleic acid counterstain (blue).

*Il18* gene expression was not significantly altered in this model (**Fig 3B**). A granular expression of IL1B was present within neuronal somata as well as an increasing expression on non-neuronal cells at all analysed timepoints with the strongest expression after 28 days in the dorsal CA1 region (**Fig 3C and 3D**). Interestingly, in the chronic period, an intense IL1B expression was present on non-neuronal cells including astrocytes (**Fig 3E**). In addition to a large inter-individual variance in seizure frequency, the median frequency of seizure sum of three consecutive days shows minor differences between the different time points showing at both models (**S3A and S3B Fig**). A direct inverse correlation between seizure frequency and gene expression was observed for Interleukin 1 beta (*Il1b*) in the KA-induced model (**S3B Fig and S2 Table**).

## Microglia and astrocytes exhibit different activation states during disease progression

Activated microglia and astrocytes have been shown to play an important role in neuroinflammation and activation of proinflammatory pathways in many CNS diseases including TLE [23, 36, 37]. Furthermore, astrogliosis is one of the most important hallmarks of HS. We investigated microglia and astrocytes using their specific marker proteins AIF1 and GFAP, respectively. After pilocarpine induced SE, a significant increase of *Aif1* (**Fig 4A**, leftmost) as well as *Gfap* (**Fig 4A**, rightmost) transcription was observed with real-time RT-PCR at all analysed time points. Accordingly, the area of IBA1- and GFAP-positive cells was also increasing over the observed time (**Fig 4B**), indicating increased cell activation. Single activated microglial cells with amoeboid shape were already present in the CA1 region 72 hours after pilocarpine-induced SE (**Fig 4C**). Microglial cells with a prominent cell body and little ramification accumulated along the hippocampal fissure (**Fig 4C**, upper panel; **S2 Fig**). At 10 days post-SE an increased number of microglia was observed, most prominently in the medial part of the CA1 cell band and the transition between CA1 and CA2 somata. Next to bushy morphology also various amoeboid shaped microglia (**Fig 4C**) were observed. Four weeks after SE, activated microglia were still present but the majority in direct proximity of CA1 pyramidal cell somata appeared more ramified (bushy) and above the hippocampal fissure with a bushy and hypertrophic shape, classified according to Wyatt-Johnson [38]. Reactive, contiguous but essentially non-overlapping astrocytes with

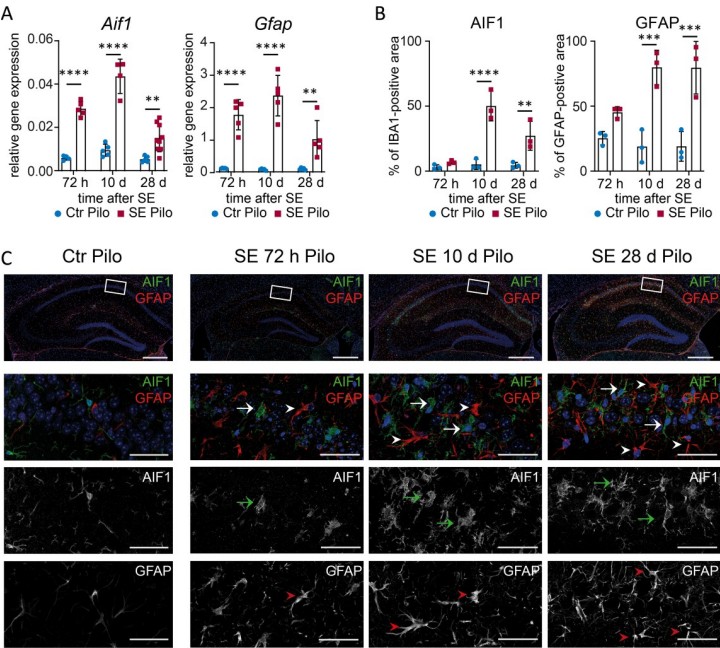

**Fig 4. Sustained microglial activation and astrogliosis after pilocarpine-induced SE.** (**A**) mRNA level of Aif1 (microglia) and Gfap (astrocytes) are analysed 72 h (n ≥ 5), 10 d (n ≥ 4) and 28 d (n ≥ 6) after pilocarpine-induced SE (SE Pilo) compared to non-SE control animals (Ctr Pilo) using the housekeeping gene β-actin (ΔΔC$_t$ method). (**B**) Semi-quantitative analysis of AIF1- or GFAP-positive area reveal a progressive increase over time in CA1 (all groups n = 3). Data is analysed with 2way ANOVA followed by Sidak's post hoc test. Asterisks indicate significant differences between groups (control vs. SE): *p < 0.05, **p < 0.01, ***p < 0.001, ****p < 0.0001. Detailed statistical values are supported in **S2 Table**. (**C**) Single AIF1 positive cells (green) can be found 72 h after SE with an increasing number 10 d and 28 d post-SE (arrows). GFAP positive astrocytes (red) with enlarged stem processes are increasingly distributed over the hippocampal section with disease progression (arrowheads). In non-SE control animals (leftmost panels, Ctr Pilo), only single ramified microglia and non-overlapping astrocytes with very fine processes can be found. Scale bars: 500 μm (overview), 50 μm (insert). DAPI for nucleic acid counterstain (blue).

enlarged stem processes were observed 72 hours and 10 days post-SE indicating a mild to moderate astrogliosis (**Fig 4C**), classified according to Sofroniew [39] (**S2 Fig**). In the chronic state (28 days after SE), abundant astrocytes were detectable, leading to an overlap of astrocytic cellular processes. However, the cell bodies and processes were less prominent, which points towards a less reactive state. In non-SE control animals, single ramified microglia and non-overlapping astrocytes with very fine processes were detected (**Fig 4C**).

In contrast to the pilocarpine model, only a transiently augmented expression of *Aif1* was present 72 hours and 10 days after disease onset (**Fig 5A**, leftmost) in the KA-model. *Gfap* mRNA was increased at 72 hours and 28 days, but not 10 days after SE (**Fig 5A**, rightmost). In addition, semi-quantitative analysis of astrocytic and microglial activation revealed an increasing volume of these cell types during the model progression in the ipsilateral CA1 region (**Fig 5B**), whereas the contralateral CA1 was mostly unaffected (**S1B Fig**). An increase in microglia was only evident contralaterally in the later model phase (28 days), possibly as a late secondary immune response. Different stages of microglia activation (ramified, hypertrophic, bushy, and amoeboid) were observed after KA-induced SE (**S2 Fig**). In contrast, rod and single ramified non-activated microglia could be found in non-SE controls (**Fig 5C**). A diffuse accumulation of activated bushy microglia was seen already 72 hours after SE in the entire ipsilateral hippocampal formation (**Fig 5C**), whereas 10 days after SE, amoeboid and bushy shaped microglia appeared in a more clustered and condensed pattern in the CA1 region. In the contralateral hippocampal formation, the microglia appeared in amoeboid shapes (**Fig 5D**). Interestingly, 10 days after SE microglia returned to a non-activated ramified morphology, as observed in non-SE controls. At 28 days post SE, a dense network of amoeboid shaped microglia was observed, clustered in CA1 and subiculum. Astrocytes (**Fig 5C**) appeared non-overlapping with hypertrophy of stem processes within the entire hippocampal area 72 hours after SE, indicating a mild to moderate astrogliosis. Ten days after SE, the number of astrocytes increased mainly in CA1 and DG of the ipsilateral hippocampus. Reactive astrocytes showed overlapping cell processes and some cell bodies exhibited corresponding signs of hypertrophy, whereas others lost individual domains, representing a severe diffuse astrogliosis [39]. Contralateral to the injection side, only a few astrocytes with slightly thickened stem processes could be observed (**Fig 5D**). Four weeks after SE, astrogliosis on the ipsilateral side spread from the respective areas (CA1 and DG) and included the entire hippocampal formation. Astrocytes in the ipsilateral CA1 region presented elongated shapes and overlapping processes, demonstrating a severe astrogliosis with compact scar formation. In the contralateral hippocampal formation, numerous hypertrophic astrocytes indicative of a severe diffuse astrogliosis were found in the CA1 region (**Fig 5D**).

## Robust expression of NLRP3 associated transcripts is independent of neuronal damage

Hippocampal biopsy specimen of TLE patients (**Fig 6A**, H. & E.) who underwent surgical removal of the affected hippocampal formation for seizure relief, demonstrated distinct neuropathological features. HS-TLE is presented with a considerably stronger structural reorganization and neuronal cell loss compared to lesion-associated TLE (**Fig 6A**, NeuN). Further immunohistochemical analysis showed an extensive astrogliosis in HS-TLE hippocampi, but only mild to moderate stages in lesion-associated TLE hippocampi (**Fig 6B**, GFAP). Microglia infiltration was observed by AIF1 immunohistochemistry in all TLE hippocampi, independently of the neuropathological phenotype (**Fig 6B**, AIF1). Additional immunolabeling against NLRP3 also revealed a similar expression pattern in both TLE pathologies evaluated by morphological analysis (**Fig 6C**).

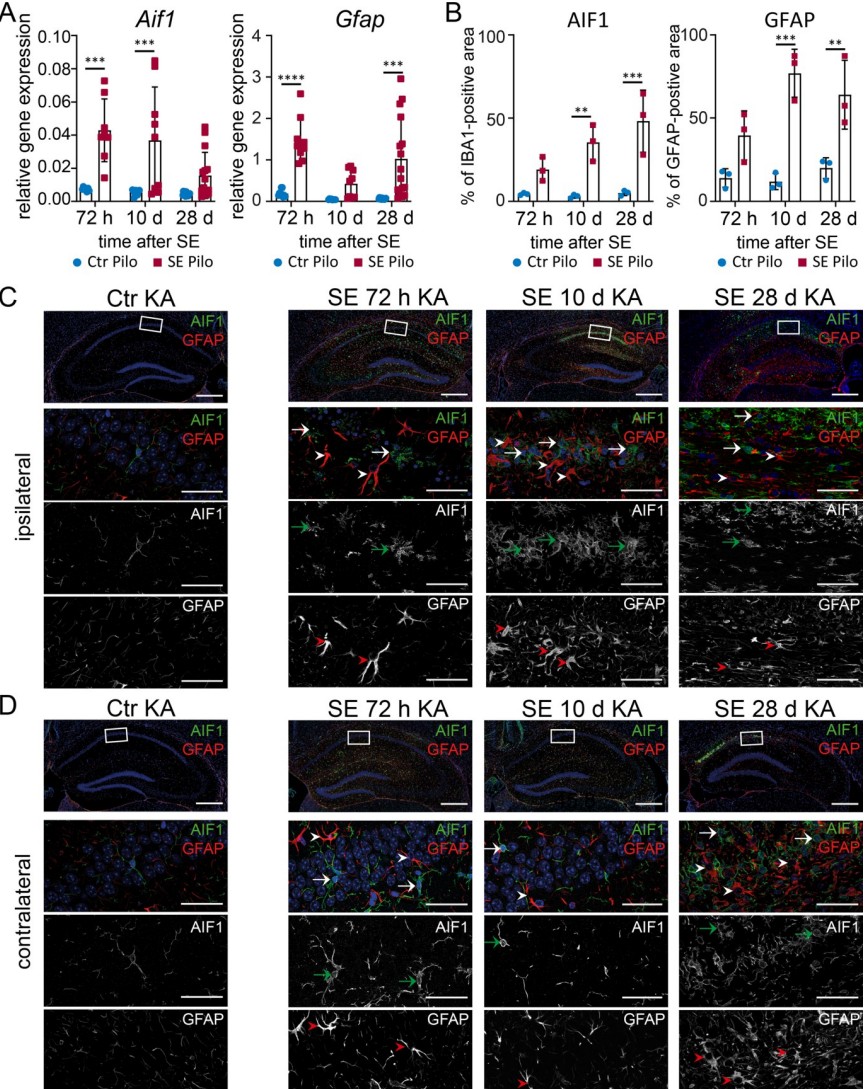

**Fig 5. Astrogliosis and microglial activation are persistent after kainic acid-induced SE.** (A) mRNA level of Aif1 and Gfap are determined 72 h (n ≥ 4), 10 d (n ≥ 5) and 28 d (n ≥ 6) after SE using the housekeeping gene β-actin (ΔΔC$_t$ method). (B) Semi-quantitative analysis of AIF1- or GFAP-positive area reveal a progressive increase over time in the ipsilateral CA1 (all groups n = 3). Data is analysed with 2way ANOVA followed by Sidak's post hoc test. Asterisks indicate significant differences between groups: *p < 0.05, **p < 0.01, ***p < 0.001, ****p < 0.0001. Detailed statistical values are supported in S2 Table. (C) AIF1 (green) positive microglial cells are manifested in the hippocampus post-SE primarily presented with a bushy morphology (arrows) becoming bushier. GFAP positive cells (red) with prominent stem processes are equally scattered over the hippocampal area (arrowheads) mainly found in the CA1 and DG region, with cell processes overlapping to a high degree. In non-SE control animals, a few ramified microglia are observed (leftmost). GFAP positive cells are similarly scarce and presented with very narrow cell processes. Scale bars: 500 μm (overview), 50 μm (insert). DAPI for nucleic acid counterstain in blue.

When proinflammatory reorganization was analysed on the cellular level by RNA sequencing (**Fig 6D**), HS-TLE tissue revealed a significantly increased GFAP expression compared to lesion-associated TLE indicating a higher number of reactive astrocytes. In contrast, accordingly to the protein expression, *Aif1* mRNA, a marker for microglia, was not differentially expressed between the two groups. Additionally, we analysed gene expression of NLRP3 pathway components. The majority of genes (namely *Nlrp3*, *Casp1*, *NfkB2*, *Il1b* and *Il18*) did not

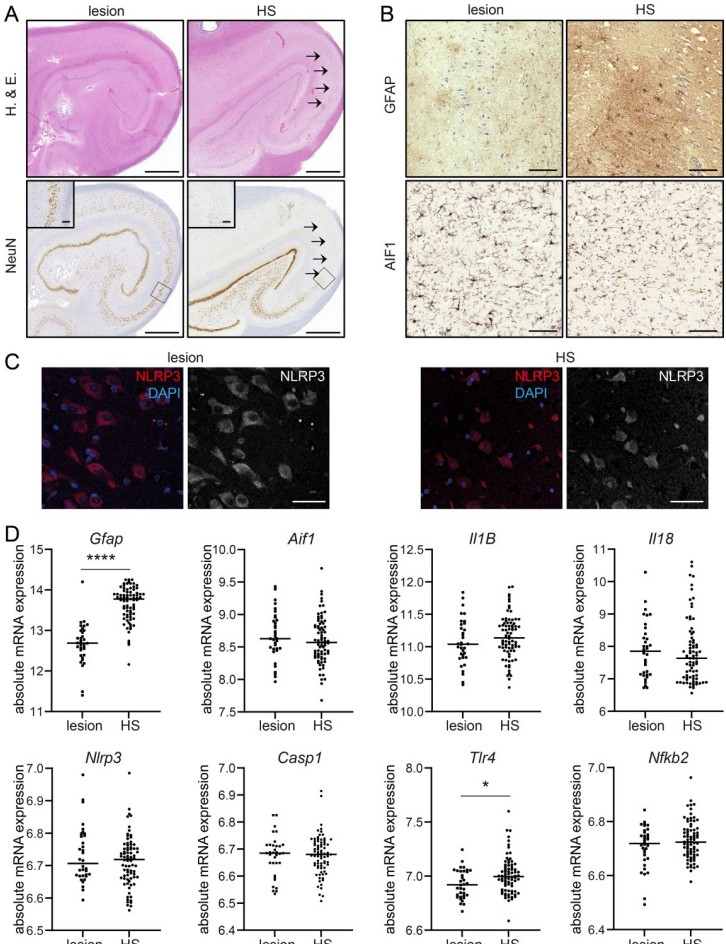

**Fig 6. Activation of NLRP3-signalling associated genes in human TLE hippocampi.** (**A**) Representative NeuN immunohistochemistry reveals a classical pattern for HS-TLE with segmental neuronal loss pronounced in CA1 (rightmost, arrows) whereas neurons are well conserved in the lesion-associated hippocampus. Scale bars: 500 μm (overview), 100 μm (insert). (**B**) Only scattered astrocytes are observed in lesion-associated TLE (leftmost, see inserts), compared with pronounced fibrillary reactive astrogliosis in HS-TLE (leftmost). Similar reactivity of activated microglia (AIF1) is observed in both pathologies (lower panels). Scale bars: 100 μm. (**C**) Representative NLRP3 immunolabeling also shows a similar expression pattern in both TLE forms. Scale bars: 50 μm. (**D**) mRNA level of *Gfap*, *Aif1*, *Il1B*, *Il18*, *Nlrp3*, *Casp1*, *Tlr4 and Nfkb2* are determined with RNA sequencing in HS-TLE (n = 78) and lesion-associated TLE patients (n = 34). Data is analysed with Mann-Whitney U test. Asterisks indicate significant differences between groups: *p < 0.05, ****p < 0.0001.

show a difference between HS and lesion-associated pathology. However, *Tlr4* mRNA was significantly augmented in HS-TLE hippocampi (**Fig 6D**).

## Discussion

Here, we have comparatively analysed the dynamics of inflammasomal pathway components after distinct transient insults. Our data suggest fundamental differences dependent of the particular stimulus, i.e., systemic pilocarpine versus suprahippocampal KA. It has to be noted that molecular activation patterns can be affected by specific pharmacological induction triggering the immediate pharmacological effects on immunological signalling. Pilocarpine acts directly on Caspase-1/IL-1β signalling, in particular at muscarinic M3 receptors and thus impacts on microglia as they express functional muscarinic receptors [40, 41]. KA has been demonstrated

to activate inflammasomal molecules via binding to Girk1 and other KA-receptors and to induce endoplasmatic reticulum stress [42]. However, the effects of two key factors–SE *and* immediate pharmacological signalling—are virtually indistinguishable in the present models. Thus, the differences between the pilocarpine- and KA-induced SE models in terms of different activation profiles of inflammasome-related molecules suggest distinct and specific dynamics of inflammasome signalling dependent on the stimulus.

Previous animal studies have shown an augmentation and activation of NLRP3 pathway proteins as an acute response to chemoconvulsant and electrically kindling-induced SE (24–72 h post SE) [12, 13, 25, 43]. We investigated the gene expression pattern over a four week period in two chemoconvulsant models, which are known to replicate the neuropathological features found most commonly in human TLE: hippocampal sclerosis with extensive neuronal death and structural reorganization [6]. In the pilocarpine model, mice show a prolonged inflammatory response, which is still present at 10 d after SE, but not in the later chronic phase (28 days). However, mice in the KA-model present with a biphasically appearing inflammatory activation, showing an augmentation of proinflammatory mRNA at 72 h post SE as well as in the later chronic model stage although seizure frequency remains stable until week 6 after SE in this model [27]. Previous animal studies have shown an augmentation and activation of NLRP3 pathway proteins as an acute response to chemoconvulsant and electrically kindling-induced SE (24–72 h post SE) [12, 13, 25, 43]. For IL1B expression, in line with previous findings after pilocarpine-induced SE in rats [44], we observed immunoreactivity at all investigated time points. We observed a shift from mainly neuronal IL1B expression at early stages (72 h post SE) to glial expression at later stages (10 d and 28 d post SE) in both models. Early neuronal IL1B expression may be a response to excitotoxic cell damage. Whereas neuronal IL1B expression in the KA model showed a broad dispersion across CA1, it appeared to be spatially clustered only in a small number of neurons in the pilocarpine model. Clustered expression of IL1B in neurons associated with TLE has been shown previously in rats after electrically induced SE within the stimulated CA3 region [45]. We hypothesize that SE induced by systemic administration of pilocarpine leads to indirect neuronal damage by hyperexcitation, and thus to a clustered pattern of damage. In contrast, local application of kainic acid above the CA1 region leads to a more widespread direct toxic effect on neurons [46] resulting in broad neuronal activation throughout the CA1 region. This may then induce a diffuse inflammation and IL1B expression in the CA1 area that is severe enough to be detectable by mRNA quantification. In contrast, in the pilocarpine model, neurons have been shown to form spatially restricted functional clusters [47] with segmental neuronal cell loss as a common hallmark [48, 49], which can be detected already 2 days after pilocarpine-induced SE in the CA1 region [48, 50].

Hippocampi after pilocarpine-induced SE as well as the ipsilateral hippocampus after suprahippocampal KA-induced SE show extensive neuronal damage. Thus, we speculated whether neurodegeneration, e.g., via release of cell death-associated molecules, may be critical in stimulating inflammasomal signalling. However, the lack of significant differences in the expression of inflammasome-associated signals between the ipsi- and contralateral (less damaged) hippocampal formation in KA-exposed mice suggest that seizure activity itself appears to be the most important factor in active inflammasomal signalling.

With respect to the role in human TLE, NLRP3 and associated proteins (CASP1, ASC, IL1B) have been shown to be upregulated in hippocampi of patients with pharmacoresistant TLE [12, 24, 25]. However, these studies have focussed on analysing inflammasome signalling in human hippocampi with the damage pattern of HS. Comparisons between expression patterns in biopsies from epilepsy surgery in pharmacoresistant patients with HS *plus* epileptogenesis with lesion-associated TLE and absent segmental neurodegeneration *without* epileptogenesis [4, 6] may provide important information about the role of pathomechanisms

for distinct aspects of human TLE [30]. Due to the general absence of matching non-epileptic human control tissue, the comparison of neuropathological and molecular properties between HS and lesion-associated human TLE is used to gain a better understanding of underlying pathogenetic mechanisms [6, 30].

Our present data clearly demonstrate that in chronic human TLE hippocampi there is robust and largely indistinguishable expression of key inflammasome components independent of both, the neurodegenerative state and the extent and type of astrogliosis. Only expression of Tlr4 was significantly increased in HS versus lesion-associated hippocampi. This finding is well in line with previous data demonstrating the importance of the HMGB1-TLR4 axis for this type of TLE pathology [11]. Our data indicate a certain level of specificity for this mechanism in HS- compared to lesion-associated TLE, which is also supported by the previously described positive correlation of Tlr4 expression levels in HS hippocampi with seizure activity [34].

Activated microglia and reactive astrocytes are associated with release of proinflammatory cytokines in human and experimental TLE [23, 36, 37, 51]. Furthermore, these cell types have been shown to contribute to epileptogenesis in experimental models [9, 52]. We analysed *Aif1* and *Gfap* mRNA as well as protein localization to determine the timeline of more general proinflammatory changes in both animal models and human chronic TLE. As to be expected, HS-TLE patients present with a stronger structural reorganization, neuronal cell loss and astrogliosis compared to lesion-associated TLE patients. However, microglia infiltration or NLRP3 expression does not differ between the two pathologies. Concordant to the time line of the more specific (NLRP3 pathway-related) findings in our experimental models, highest mRNA expression and more 'activated' cell morphology for both microglia and astrocytes is found at 10 days post SE in the pilocarpine model. In the later chronic phase, densely packed and highly activated microglia and astrocytes along with increased *Gfap* and *Aif1* mRNA levels are found in the KA-model. A study focussing on even later time points in the suprahippocampal KA-model shows that astrocyte-related pathogenic changes are progressing up to nine month post-SE [9]. The time lag between mRNA generation and protein expression at day 10 in the KA model might be due to the highly variable and complex translational processes [53]. Post-translational modifications, such as phosphorylation, which important for protein stability and function, also happen on different time scales [54]. Consequently, the expression of both, mRNA and protein, may vary at the same time point.

Thus, an important finding arising from this study is a potentially distinct mechanism causing diverse time courses of inflammatory activity in the two different mouse models with similar seizure frequency during various model stages [9, 28, 29, 55]. The biphasic inflammatory timeline in the KA model could be attributed initially to an acute response to excitotoxicity and SE, with an ongoing chronic progression of inflammation and cellular remodelling, eventually leading to destruction of the hippocampal formation. Potentially, these chronic pathogenetic changes could be caused by self-perpetuating inflammatory cascades (such as the NLRP3 pathway). Another possibility is an association of IL1B driven chronic neuroinflammation with increased seizure frequency. Different studies report a progression of seizure frequency in the KA-model [9, 56, 57] which we could confirm. In contrast, a lasting increase in seizure frequency in the pilocarpine model is detected just for subgroups of animals [7, 58] with low but stable seizure occurrence in the later phase [28, 29]. It has been shown that IL1B contributes to hyperexcitability [59], which gives rise to the hypothesis that inflammatory processes in the chronic state after KA- induced SE could promote an increase in seizure frequency. Although a direct correlation between seizure frequency and gene expression was only found for Il1b in the KA model, this does not argue against a dependence of both parameters. Seizure frequency is extremely variable inter-individually, so a comprehensive transcriptional analysis with

individual seizure frequency should be considered for a solid analysis. Moreover, it is also possible that the changes in the expression of the NLRP3-associated genes are secondary phenomena of the epileptogenic process, for instance, due to altered cell composition or cytokine accumulation in brain tissue, rather than a direct consequence of altered neuronal balance.

In summary, our data from human and experimental TLE hippocampi suggest that inflammasome pathway activation and seizure activity are inextricably linked in the manifestation of a chronic epileptic state. Given the highly dynamic epileptogenic transcript profiles that depend on the particular insult modalities, further research is needed to address highly specific, distinct molecular components of inflammasome signalling in a therapeutic context for pharmacoresistant TLE.

## Supporting information

**S1 Fig. No difference in mRNA expression of inflammasomal related genes is observed between ipsi- and contralateral CA1 region after KA-induced SE.** (**A**) Relative mRNA levels ($\Delta\Delta C_t$ method) are analysed 72 h (n $\geq$ 4), 5 d (n $\geq$ 5), 10 d (n $\geq$ 4) and 28 d (n $\geq$ 6) after kainic acid-induced SE comparing ipsi- and contralateral CA1 with 2way ANOVA followed by Sidak's post hoc test. The ubiquitously expressed *β-actin* is used as housekeeping gene. (**B**) Semi-quantitative analysis of AIF1- or GFAP-positive area reveal nearly no significant changes over time in the contralateral CA1 (all groups n = 3) with 2way ANOVA followed by Sidak's post hoc test. An increase in microglia (AIF1) was only evident contralaterally in the later model phase (28 days). Asterisks indicate significant differences between groups: $^*$p < 0.05. Detailed statistical values are found in **S2 Table**.
(TIF)

**S2 Fig. Different cell shapes of microglia and astrocytes.** (**A**) Visualisation of AIF1 positive microglia (green) and (**B**) GFAP positive astrocytes (red) presenting different shapes of soma and projections depending on the activation status after SE. Scale bars: 20 μm.
(TIF)

**S3 Fig. Almost no correlation between seizure frequency and gene expression.** (A, B) Median of the summarized seizure frequency of three consecutive days before mRNA analysis for every animal after (**A**) pilocarpine (n = 11) and (**B**) KA-induced SE (n = 21) correlated to mRNA expression level of representative genes. (**C, D**) Simple regression analysis of seizure frequency and mRNA gene expression of three representative genes analysed after (**C**) pilocarpine and (**D**) KA-induced SE. Asterisks indicate significant differences between groups: $^*$p < 0.05. Detailed statistical values are found in **S2 Table**.
(TIF)

**S1 Table. Distribution of clinical parameters with the "lesion associated" and hippocampal sclerosis (HS) patient groups.** LEV = Levetiracetam. Engel class IA: completely seizure free; class IVB: no seizure freedom.
(PDF)

**S2 Table. Statistical analysis.**
(PDF)

## Acknowledgments

We thank Pia Trebing and Sabine Opitz for excellent technical assistance. The Department of Epileptology is a full member of the ERN EpiCARE.

## Author Contributions

**Conceptualization:** Thoralf Opitz, Rainer Surges, Motaz Hamed, Hartmut Vatter, Susanne Schoch, Albert J. Becker, Julika Pitsch.

**Data curation:** Malin S. Pohlentz, Silvia Cases-Cunillera.

**Formal analysis:** Malin S. Pohlentz, Silvia Cases-Cunillera, Thoralf Opitz, Susanne Schoch, Julika Pitsch.

**Funding acquisition:** Susanne Schoch, Albert J. Becker, Julika Pitsch.

**Investigation:** Philipp Müller, Albert J. Becker, Julika Pitsch.

**Methodology:** Philipp Müller, Motaz Hamed, Hartmut Vatter, Susanne Schoch.

**Project administration:** Albert J. Becker, Julika Pitsch.

**Resources:** Motaz Hamed, Hartmut Vatter.

**Supervision:** Albert J. Becker, Julika Pitsch.

**Validation:** Philipp Müller, Rainer Surges, Susanne Schoch, Julika Pitsch.

**Visualization:** Malin S. Pohlentz, Silvia Cases-Cunillera, Julika Pitsch.

**Writing – original draft:** Malin S. Pohlentz, Silvia Cases-Cunillera, Thoralf Opitz, Susanne Schoch, Albert J. Becker, Julika Pitsch.

**Writing – review & editing:** Malin S. Pohlentz, Thoralf Opitz, Rainer Surges, Motaz Hamed, Hartmut Vatter, Susanne Schoch, Albert J. Becker, Julika Pitsch.

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
