## [Decision Letter · Decision Letter 0]

14 Dec 2021

PONE-D-21-35199Characterisation of NLRP3 pathway-related neuroinflammation in temporal lobe epilepsyPLOS ONE

Dear Dr. Pitsch,

Thank you for submitting your manuscript to PLOS ONE. After careful consideration by 2 Reviewers and an Academic Editor, all of the critiques of both Reviewers must be addressed in detail in a revision to determine publication status. If you are prepared to undertake the work required, I would be pleased to reconsider my decision, but revision of the original submission without directly addressing the critiques of the Reviewers does not guarantee acceptance for publication in PLOS ONE. If the authors do not feel that the queries can be addressed, please consider submitting to another publication medium. A revised submission will be sent out for re-review. The authors are urged to have the manuscript given a hard copyedit for syntax and grammar.

Journal Requirements:

2.We note that the grant information you provided in the ‘Funding Information’ and ‘Financial Disclosure’ sections do not match.

 (Our work is supported by the Deutsche Forschungsgemeinschaft (to AJB: SFB 1089; to SS: SCHO 820/4-1, SCHO 820/6-1, SCHO 820/7-1, SCHO 820/5-2, SPP1757, SFB1089; to AJB: FOR 2715), Else Kröner-Fresenius-Foundation (Promotionskolleg ‘NeuroImmunology’ to MP, SS, AJB; 2016_A05 to JP), as well as the BONFOR program of the Medical Faculty, University of Bonn (MP, SS, AJB, JP). The funders had no role in study design, data collection and analysis, decision to publish, or preparation of the manuscript.)

(Funding: Our work is supported by the Deutsche Forschungsgemeinschaft (to AJB: SFB 1089; to SS: SCHO 820/4-1, SCHO 820/6-1, SCHO 820/7-1, SCHO 820/5-2, SPP1757, SFB1089; to AJB: FOR 2715), Else Kröner-Fresenius-Foundation (Promotionskolleg ‘NeuroImmunology’ to MP, SS, AJB; 2016_A05 to JP), as well as the BONFOR program of the Medical Faculty, University of Bonn (MP, SS, AJB, JP). The Department of Epileptology is a full member of the ERN EpiCARE. The funders had no role in study design, data collection and analysis, decision to publish, or preparation of the manuscript. The present work contributes to the requirements for MP’s written thesis to obtain the MD degree at the Medical Faculty, University of Bonn. )

(Our work is supported by the Deutsche Forschungsgemeinschaft (to AJB: SFB 1089; to SS: SCHO 820/4-1, SCHO 820/6-1, SCHO 820/7-1, SCHO 820/5-2, SPP1757, SFB1089; to AJB: FOR 2715), Else Kröner-Fresenius-Foundation (Promotionskolleg ‘NeuroImmunology’ to MP, SS, AJB; 2016_A05 to JP), as well as the BONFOR program of the Medical Faculty, University of Bonn (MP, SS, AJB, JP). The funders had no role in study design, data collection and analysis, decision to publish, or preparation of the manuscript.)

Funding: Our work is supported by the Deutsche Forschungsgemeinschaft (to AJB: SFB 1089; to SS: SCHO 820/4-1, SCHO 820/6-1, SCHO 820/7-1, SCHO 820/5-2, SPP1757, SFB1089; to AJB: FOR 2715), Else Kröner-Fresenius-Foundation (Promotionskolleg ‘NeuroImmunology’ to MP, SS, AJB; 2016_A05 to JP), as well as the BONFOR program of the Medical Faculty, University of Bonn (MP, SS, AJB, JP). The Department of Epileptology is a full member of the ERN EpiCARE. The funders had no role in study design, data collection and analysis, decision to publish, or preparation of the manuscript. The present work contributes to the requirements for MP’s written thesis to obtain the MD degree at the Medical Faculty, University of Bonn.

**Comments to the Author**

1. Is the manuscript technically sound, and do the data support the conclusions?

Reviewer #1: Partly

Reviewer #2: Partly

2. Has the statistical analysis been performed appropriately and rigorously? 

Reviewer #1: Yes

Reviewer #2: Yes

3. Have the authors made all data underlying the findings in their manuscript fully available?

Reviewer #1: Yes

Reviewer #2: Yes

4. Is the manuscript presented in an intelligible fashion and written in standard English?

Reviewer #1: Yes

Reviewer #2: Yes

5. Review Comments to the Author

Reviewer #1: The manuscript written by Pohlentz et al. focuses on the process of neuroinflammation in different models of epilepsy and in human epileptic tissues. It suggests, not for the first time, the dynamic activation of the NLRP3 inflammosome following chemoconvulsive administrations in animal models and in human hippocampi of TLE. The NLRP3 inflammasome and related signaling molecules have been extensively analyzed at different time points in the epileptic process, both as transcripts and as proteins, concluding that there are specific and distinct dynamics of inflammasome signaling, dependent on the stimulus.

Overall, I would like the authors to revisit a few points:

- models of epilepsy are not well characterized: more information is needed on the frequency and severity of seizures, the percentage of animals that do not respond or die, and the progression of the epileptic process. These data could be very useful to the authors for a more in-depth discussion of the expression patterns of inflammasome components at different time points in epileptogenesis. It should be clarified for example whether 10 days after chemoconvulsant treatment, the animals are in the latency period (without recurrent spontaneous seizures) or in chronic epilepsy. In the discussion, the authors mention that the two animal models (pg 18- second paragraph) have a similar seizure frequency in the chronic period, but they report no confirmatory data.

- general clinical information regarding TLE patients should be provided: adding age, sex, time of epilepsy diagnosis, seizure frequency, drug treatment, the authors could present a more complete characterization of tissue samples.

- some findings should be better addressed and discussed:

Why does IL1beta protein expression increase at 72 h postSE in the pilocarpine model whereas mRNA at 10 days? Furthermore, the authors observed IL1beta signals in a clustered group of neurons: does this evidence find confirmation in the literature?

In the kainate model, GFAP mRNA and protein show a different time course: why at 10 days GFAP mRNA does not increase while many GFAP positive cells are present on ipsilateral tissue slices?

Minor points:

-page 6 in the induction of chronic epilepsy by systematic injection of pilocarpine, please specify which animals were used.

- Data are represented with different graphical forms: the best layout is the one as scatterplot and should be used also in fig.1, 2 and 3.

Reviewer #2: The manuscript is well written and the data are interesting and consistent. However, some concerns should be approached:

1. Please, avoid complex/confuse sentences (e.g. the last paragraph of the introduction section should be rewritten).

2. The statistical tests applied and their goals should be described in the material and methods section. The suppl. material only presents the analysis results, but not describe the statistical methods properly.

3. The authors affirm that it was observed an increased number of microglial and astroglial cells (main text and Figs. 4 and 5). However, they did not perform a morphometric analysis (e.g. fluorescence intensity of AIF1 and GFAP or positive cell count) to confirm this statement.

4. According to the authors, "additional immunolabeling against NLRP3 also revealed a similar expression pattern in both TLE pathologies". How was this expression pattern evaluated (morphological and/or morphometric analysis)? Please, include this information in the materials and methods section.

5. The authors affirm that "In the chronic phase, densely packed and highly activated microglia and astrocytes along with increased Gfap and Aif1 mRNA level are found in the KA-model". Although the number of microglia seemed increased in KA-model compared to control group at 28 days, the Aif1 mRNA level was not. Once again, the morphometric analysis of AIF1 expression should be useful to complement the mRNA findings.

6. PLOS authors have the option to publish the peer review history of their article (what does this mean?). If published, this will include your full peer review and any attached files.

**Do you want your identity to be public for this peer review?** For information about this choice, including consent withdrawal, please see our Privacy Policy.

Reviewer #1: No

Reviewer #2: No

We look forward to receiving your revised manuscript.

Kind regards,

Stephen D. Ginsberg, Ph.D.

Section Editor

PLOS ONE

---

## [Author Response · Author response to Decision Letter 0]

13 Jun 2022

Responses to the individual referees’ comments on the manuscript entitled "Characterisation of NLRP3 pathway-related neuroinflammation in temporal lobe epilepsy” (Manuscript ID: PONE-D-21-35199)

Editor

Point 1: The authors are urged to have the manuscript given a hard copyedit for syntax and grammar.

Response: The entire revised document was read by a native speaker and corrected for syntax, grammar and spelling.

Reviewer 1.

The individual points of the reviewer were addressed in detail as follows:

Point 1: models of epilepsy are not well characterized: more information is needed on the frequency and severity of seizures, the percentage of animals that do not respond or die, and the progression of the epileptic process. These data could be very useful to the authors for a more in-depth discussion of the expression patterns of inflammasome components at different time points in epileptogenesis.

It should be clarified for example whether 10 days after chemoconvulsant treatment, the animals are in the latency period (without recurrent spontaneous seizures) or in chronic epilepsy. In the discussion, the authors mention that the two animal models (pg 18- second paragraph) have a similar seizure frequency in the chronic period, but they report no confirmatory data.

Response: We thank the referee for this comment. The reviewer is correct in the notion that we did not report on extensive details of the epilepsy phenotype of the models used in our study. We have limited this information, since it would be a pure repetition of published data. In fact, both epilepsy models used in our study are very well established and there is an extensive body of literature on their characterization. We believe that a key novelty aspect of the present manuscript is given by the expression patterns of inflammasome components; we discuss those with regard to these published data. The definition of a latency (“silent”) intervals in the epileptogenesis models may represent an own independent topic (Becker, 2018; Levesque et al., 2021). In fact, 24/7 EEG deep electrode measurements accompanied with video monitoring reveals ongoing seizure activity even in the days immediately after chemoconvulsant treatment (Mazzuferi et al., 2012; Pitsch et al., 2017). Considering these aspects, we hope that the referee is in line with our preference to refer to time points after SE induction in days rather than in model ‘stages’, which may insinuate already interpretational aspects that are not directly related to the topic of this manuscript. These considerations have been included in the manuscript in a condense fashion and we have added two additional references (Bedner et al., 2015; Curia et al., 2008) that also describe key data on the seizure pattern of both models in great detail (p. 20, l. 26 – p. 21, l. 1).

Point 2: general clinical information regarding TLE patients should be provided: adding age, sex, time of epilepsy diagnosis, seizure frequency, drug treatment, the authors could present a more complete characterization of tissue samples.

Response: According the reviewers’ suggestion, we have now added a table with the general clinical information in the Supplement (Suppl. table 1). 

Hippocampal biopsies from patients with chronic pharmacoresistant mesial TLE were used for the present analysis (Wiebe et al., 2001). In all patients, preoperative examination with a combination of non-invasive and invasive procedures revealed that the seizures originated in the mesial temporal lobe (Kral et al., 2002). Surgical resection of the hippocampus was clinically indicated in all cases because of pharmacoresistance. Hippocampal tissue samples were available for neuropathologic studies for each case included in the present study. The hippocampal sclerosis (HS) group was clearly characterized by segmental neuronal cell loss and concomitant astrogliosis and microglial activation. The hippocampi in the control group showed no segmental neuronal cell loss neuropathologically but had astrogliosis and microglial activation and were therefore consistent with lesions such as cortical dysplasia or epilepsy-associated tumors. In each case, the diagnosis was made by an experienced neuropathologist (AJB) according to international criteria (Becker et al., 2003; Blümcke et al., 2007). We added this additional information in the revised manuscript (p. 9, l. 17- p10, l. 1).

Point 3: some findings should be better addressed and discussed: Why does IL1beta protein expression increase at 72 h post SE in the pilocarpine model whereas mRNA at 10 days? Furthermore, the authors observed IL1beta signals in a clustered group of neurons. Does this evidence find confirmation in the literature?

Response: At 72 h after pilocarpine-induced SE, we see IL1-beta protein expression clustered only in a small number of neurons. To quantify mRNA expression, we analyzed a large proportion of the dorsal CA1 region. Thus, the locally increased IL1-beta expression 72 h post SE may have been below the detection limit due to a dilution effect. Clustered expression of IL1 beta in neurons associated with TLE has been shown previously in rats after electrically induced SE within the stimulated CA3 region (De Simoni et al., 2000). Segmental neuronal cell loss in CA1 is a common hallmark after pilocarpine-induced SE in chronic rodents (Becker et al., 2008; Mello et al., 1993). However, using TUNEL or silver staining, neuronal cell death was detected in a clustered group of neurons in wildtype mice already 2 days after pilocarpine-induced SE in the CA1 region (Becker et al., 2008; Covolan and Mello, 2000). SE induced by systemic administration of pilocarpine is generally regarded to induce primarily indirect neuronal damage by hyperexcitation, and thus to a clustered pattern of damage. In contrast, local application of kainic acid above the CA1 region preferentially acts by widespread excitotoxic effects on neurons (Vezzani et al., 1999), which then induces a diffuse inflammation and IL1 beta expression in the CA1 area that is severe enough to be detectable by mRNA quantification. We have included relevant aspects in the manuscript (pp. 18, ll. 11-24).

Point 4: In the kainate model, GFAP mRNA and protein show a different time course: why at 10 days GFAP mRNA does not increase while many GFAP positive cells are present on ipsilateral tissue slices?

Response: For analyzing mRNA levels, we used the relative quantification ΔΔct-method (Becker et al., 2008; Fink et al., 1998). Expression levels of mRNA transcripts of the gene of interest are normalised to a reference gene, which is not affected by the experiment. For this purpose, we used the ubiquitously expressed gene beta actin (Actb), which is known to be stably expressed in the time course of induced epilepsy models (Chen et al., 2001; Marques et al., 2013; Pernot et al., 2010). The assessment of mRNA expression of Gfap is relative to the expression level of the reference gene Actb and is therefore also dependent on the change in total cell mass. Therefore, increased protein expression as evident in immunohistochemical analyses may not directly be translatable into mRNA quantitative measures, since e.g. dilution effects due to mixtures of cell input may interfere. In addition, there may be a time lag between mRNA generation and protein expression as the translational process is highly variable and complex (Ingolia et al., 2011). Post-translational modification, such as phosphorylation, which are known to allow the protein to function, also take different length of times (Ramazi and Zahiri, 2021). Thus, it is a common phenomenon that mRNA and protein expression for a particular gene of interest are not immediately interchangeable.

The variance between two groups was analyzed as a function of time using a 2-way ANOVA. This resulted in a p-value of 0.0089 for time variance and a p<0.0001 for group variance indicating that both groups are strongly significant different with respect to the analyzed time points and also the time point has a major impact. As pairwise comparisons in post-hoc tests are based on fewer cases than analysis of variance, this reduces the sensitivity. Therefore, the 10d time point does not appear statistically significant using the less sensitive post-hoc test, although it does show an increase when considered individually, but the difference in the other time points is also accounted for in this type of calculation.

Therefore, a combination of both methods, immunohistochemistry and quantitative mRNA analysis, is essential in order to make a meaningful statement of the disease pattern. We have included this additional information in the revised manuscript (p. 20, ll. 18-23).

Point 5: page 6 in the induction of chronic epilepsy by systematic injection of pilocarpine, please specify which animals were used.

Response: The same animals (male C57Bl6/N mice; Charles River; ~60 days old, weight ≥ 20 g) were used for both induction models. We have included this additional information in the revised manuscript (p. 6, ll. 21-22).

Point 6: Data are represented with different graphical forms: the best layout is the one as scatterplot and should be used also in fig.1, 2 and 3.

Response: We thank the reviewer for this helpful comment. According to his/her suggestion we have now adjusted the relevant figures.

Reviewer 2.

We thank the referee for commending that ‘The manuscript is well written and the data are interesting and consistent’. We particularly thank this referee for the constructive suggestion to change the presentation of our present data to a regular research article format. We feel that the manuscript has substantially benefitted from following this advice.

Point 1: Please, avoid complex/confuse sentences (e.g. the last paragraph of the introduction section should be rewritten).

Response: We thank the reviewer for this comment. We corrected the referring sentence in the revised version of the manuscript (p. 3, ll. 15-19). We also looked through the entire text and simplified all the longer sentences.

Point 2: The statistical tests applied and their goals should be described in the material and methods section. The suppl. material only presents the analysis results, but not describe the statistical methods properly.

Response: We now included the information on the applied statistical tests in the Material and Methods section (p. 11, ll. 10-16).

Point 3: The authors affirm that it was observed an increased number of microglial and astroglial cells (main text and Figs. 4 and 5). However, they did not perform a morphometric analysis (e.g. fluorescence intensity of AIF1 and GFAP or positive cell count) to confirm this statement.

Response: We addressed the Reviewer’s idea and performed further studies to quantify the morphology of microglia and astrocytes. As suggested, we now included a quantitative analysis by measuring the positively-stained area of AIF1 and GFAP to confirm our statement. Semi-quantitative analysis now underlines the picture of activation of GFAP-positive astrocytes and IBA1-positive microglia seen in the histological overview. In both models, the hippocampal formation shows an increased level of activated cells analyzed by measuring the area of AIF1- or GFAP-positive cells. The new data can now be found in Fig 4B and 5B.

Point 4: According to the authors, "additional immunolabeling against NLRP3 also revealed a similar expression pattern in both TLE pathologies". How was this expression pattern evaluated (morphological and/or morphometric analysis)? Please, include this information in the materials and methods section.

Response: We added the information that the expression pattern was evaluated by morphological analysis (p. 11, ll. 4-5 and p. 16, l. 10-11).

Point 5: The authors affirm that "In the chronic phase, densely packed and highly activated microglia and astrocytes along with increased Gfap and Aif1 mRNA level are found in the KA-model". Although the number of microglia seemed increased in KA-model compared to control group at 28 days, the Aif1 mRNA level was not. Once again, the morphometric analysis of AIF1 expression should be useful to complement the mRNA findings.

Response: Please see our answer to Point 3.

References

Becker, A.J., et al., 2003. Correlated stage- and subfield-associated hippocampal gene expression patterns in experimental and human temporal lobe epilepsy. Eur J Neurosci. 18, 2792-802.

Becker, A.J., et al., 2008. Transcriptional upregulation of Cav3.2 mediates epileptogenesis in the pilocarpine model of epilepsy. J Neurosci. 28, 13341-53.

Becker, A.J., 2018. Review: Animal models of acquired epilepsy: insights into mechanisms of human epileptogenesis. Neuropathol Appl Neurobiol. 44, 112-129.

Bedner, P., et al., 2015. Astrocyte uncoupling as a cause of human temporal lobe epilepsy. Brain. 138, 1208-22.

Blümcke, I., et al., 2007. A new clinico-pathological classification system for mesial temporal sclerosis. Acta Neuropathol. 113, 235-44.

Chen, J., et al., 2001. Activity-induced expression of common reference genes in individual cns neurons. Lab Invest. 81, 913-6.

Covolan, L., Mello, L.E., 2000. Temporal profile of neuronal injury following pilocarpine or kainic acid-induced status epilepticus. Epilepsy Res. 39, 133-52.

Curia, G., et al., 2008. The pilocarpine model of temporal lobe epilepsy. J Neurosci Methods. 172, 143-57.

De Simoni, M.G., et al., 2000. 1 Eur J Neurosci. 12, 2623-33.

Fink, L., et al., 1998. Real-time quantitative RT-PCR after laser-assisted cell picking. Nat Med. 4, 1329-33.

Ingolia, N.T., Lareau, L.F., Weissman, J.S., 2011. Ribosome profiling of mouse embryonic stem cells reveals the complexity and dynamics of mammalian proteomes. Cell. 147, 789-802.

Kral, T., et al., 2002. Preoperative evaluation for epilepsy surgery (Bonn Algorithm). Zentralbl Neurochir. 63, 106-10.

Levesque, M., et al., 2021. The pilocarpine model of mesial temporal lobe epilepsy: Over one decade later, with more rodent species and new investigative approaches. Neurosci Biobehav Rev. 130, 274-291.

Marques, T.E., et al., 2013. Validation of suitable reference genes for expression studies in different pilocarpine-induced models of mesial temporal lobe epilepsy. PLoS One. 8, e71892.

Mazzuferi, M., et al., 2012. Rapid epileptogenesis in the mouse pilocarpine model: video-EEG, pharmacokinetic and histopathological characterization. Exp Neurol. 238, 156-67.

Mello, L.E., et al., 1993. Circuit mechanisms of seizures in the pilocarpine model of chronic epilepsy: cell loss and mossy fiber sprouting. Epilepsia. 34, 985-95.

Pernot, F., et al., 2010. Selection of reference genes for real-time quantitative reverse transcription-polymerase chain reaction in hippocampal structure in a murine model of temporal lobe epilepsy with focal seizures. J Neurosci Res. 88, 1000-8.

Pitsch, J., et al., 2017. Circadian clustering of spontaneous epileptic seizures emerges after pilocarpine-induced status epilepticus. Epilepsia. 58, 1-13.

Ramazi, S., Zahiri, J., 2021. Posttranslational modifications in proteins: resources, tools and prediction methods. Database (Oxford). 2021.

Vezzani, A., et al., 1999. Interleukin-1beta immunoreactivity and microglia are enhanced in the rat hippocampus by focal kainate application: functional evidence for enhancement of electrographic seizures. J Neurosci. 19, 5054-65.

Wiebe, S., et al., 2001. A randomized, controlled trial of surgery for temporal-lobe epilepsy. N Engl J Med. 345, 311-8.

---

## [Decision Letter · Decision Letter 1]

23 Jun 2022

PONE-D-21-35199R1Characterisation of NLRP3 pathway-related neuroinflammation in temporal lobe epilepsyPLOS ONE

Dear Dr. Pitsch,

Thank you for resubmitting your work to PLOS ONE. Please make the corrections posed by Reviewer #1 so I can render a decision on this manuscript.

**Comments to the Author**

1. If the authors have adequately addressed your comments raised in a previous round of review and you feel that this manuscript is now acceptable for publication, you may indicate that here to bypass the “Comments to the Author” section, enter your conflict of interest statement in the “Confidential to Editor” section, and submit your "Accept" recommendation.

Reviewer #1: (No Response)

2. Is the manuscript technically sound, and do the data support the conclusions?

Reviewer #1: Yes

3. Has the statistical analysis been performed appropriately and rigorously? 

Reviewer #1: Yes

4. Have the authors made all data underlying the findings in their manuscript fully available?

Reviewer #1: Yes

5. Is the manuscript presented in an intelligible fashion and written in standard English?

Reviewer #1: Yes

6. Review Comments to the Author

Reviewer #1: My only concerns are still related to point 1.

I agree with the authors that:

-the characterization of the epilepsy models used in their study is fully available in the literature;

- the latency period during the development of epilepsy may not be seizure-free and is difficult to study in depth.

Despite these premises, epilepsy is a chronic and progressive disease, and in the experimental models used, several moments can be identified that differ in frequency and severity of seizures. Distinct neuroinflammation-related NLRP3 pathways are induced by SE and are modulated during the development of epilepsy (e.g., at 5-10-28 days...). The study and correlation of seizures with the dynamics of NLRP3-dependent transcripts and proteins would be of great interest.

In addition, in one part of the Discussion (p.21 l 6-12 ) there is a paragraph with a hypothesis linking seizure frequency to neuroinflammation. Hypothesis that could be supported and discussed extensively with the authors' findings.

7. PLOS authors have the option to publish the peer review history of their article (what does this mean?). If published, this will include your full peer review and any attached files.

**Do you want your identity to be public for this peer review?** For information about this choice, including consent withdrawal, please see our Privacy Policy.

Reviewer #1: No

We look forward to receiving your revised manuscript.

Kind regards,

Stephen D. Ginsberg, Ph.D.

Section Editor

PLOS ONE
---

## [Author Response · Author response to Decision Letter 1]

7 Jul 2022

Responses to the individual referees’ comments on the manuscript entitled "Characterisation of NLRP3 pathway-related neuroinflammation in temporal lobe epilepsy” (Manuscript ID: PONE-D-21-35199)

Reviewer 1.

The individual points of the reviewer were addressed in detail as follows:

Point 1: The study and correlation of seizures with the dynamics of NLRP3-dependent transcripts and proteins would be of great interest.

Response: According to the referee’s suggestions, we have added highly detailed phenotypic seizure data to the study and calculated the correlation between seizure frequency and gene expression in both models for molecules with representative expression profile changes over the time course of the epilepsy models. A direct correlation between neuronal hyperexcitability (seizure frequency) and gene expression was observed for Il1b in the KA model. Other NLRP3-related genes under study did not show similar expression changes (Representative calculation see S3 Fig and S2 Table). Direct comparability of seizure frequency, mRNA, and protein expression may be affected by time intervals between mRNA generation and protein expression in a gene-dependent manner, as the translation process is highly variable and complex (Ingolia et al., 2011). 

For correlation analysis we used the median of the summarized seizure frequency of three consecutive days before mRNA analysis for every animal (day 3: seizure sum of day 1-3; day 5: sum of day 3-5; day10: seizure sum of day 8-10; day 28: sum of day 26-28). The median seizure frequency is now found in Suppl. Fig. 3A correlated to representative genes (NLRP3, TLR4, Il1b). Correlation of these three representative genes is presented in Suppl. Fig. 3B. The statistical analysis of the correlation for all genes under study is found in Suppl. Table 2.

Point 2: In addition, in one part of the Discussion (p.21 l. 6-12) there is a paragraph with a hypothesis linking seizure frequency to neuroinflammation. Hypothesis that could be supported and discussed extensively with the authors' findings.

Response: The referee addresses an important point. Our present data suggest that robust positive correlations between expression of the inflammation associated genes under study and seizure frequencies are not evident for these molecules. Thus, our study does not support immediate conclusions such as seizure activity induced inflammation or vice versa. However, given time delays in such interfering mechanisms, we can also not rule out that an interplay between seizure activity and levels of innate inflammation exist in the models under study. To study this more in detail will require selectively interfering functionally with individual molecules under study. However, such approaches clearly go beyond the framework of the present study. We discuss this issue in the revised manuscript (p. 21, l. 22 – p. 22, l. 4).

---

## [Editor Report · Decision Letter 2]

12 Jul 2022

Characterisation of NLRP3 pathway-related neuroinflammation in temporal lobe epilepsy

PONE-D-21-35199R2

Dear Dr. Pitsch,

We’re pleased to inform you that your manuscript has been judged scientifically suitable for publication and will be formally accepted for publication once it meets all outstanding technical requirements.

Kind regards,

Stephen D. Ginsberg, Ph.D.

Section Editor

PLOS ONE

---

## [Editor Report · Acceptance letter]

5 Aug 2022

PONE-D-21-35199R2 

Characterisation of NLRP3 pathway-related neuroinflammation in temporal lobe epilepsy 

Dear Dr. Pitsch:

I'm pleased to inform you that your manuscript has been deemed suitable for publication in PLOS ONE. Congratulations! Your manuscript is now with our production department. 

Kind regards, 

on behalf of

Dr. Stephen D. Ginsberg 

Section Editor

PLOS ONE